# Harnessing Immunity to Treat Advanced Thyroid Cancer

**DOI:** 10.3390/vaccines12010045

**Published:** 2023-12-30

**Authors:** Hiroki Komatsuda, Michihisa Kono, Risa Wakisaka, Ryosuke Sato, Takahiro Inoue, Takumi Kumai, Miki Takahara

**Affiliations:** 1Department of Otolaryngology-Head and Neck Surgery, Asahikawa Medical University, Asahikawa 078-8510, Japan; komatsuda@asahikawa-med.ac.jp (H.K.); mkono@asahikawa-med.ac.jp (M.K.); r-wakisaka@asahikawa-med.ac.jp (R.W.); rsato@asahikawa-med.ac.jp (R.S.); takapiro9242@gmail.com (T.I.); miki@asahikawa-med.ac.jp (M.T.); 2Department of Medical Oncology, Dana-Farber Cancer Institute, Boston, MA 02215, USA; 3Department of Innovative Head & Neck Cancer Research and Treatment, Asahikawa Medical University, Asahikawa 078-8510, Japan

**Keywords:** thyroid cancer, immunotherapy, adjuvant, targeted therapy, peptide vaccine

## Abstract

The incidence of thyroid cancer (TC) has increased over the past 30 years. Although differentiated thyroid cancer (DTC) has a good prognosis in most patients undergoing total thyroidectomy followed by radioiodine therapy (RAI), 5–10% of patients develop metastasis. Anaplastic thyroid cancer (ATC) has a low survival rate and few effective treatments have been available to date. Recently, tyrosine kinase inhibitors (TKIs) have been successfully applied to RAI-resistant or non-responsive TC to suppress the disease. However, TC eventually develops resistance to TKIs. Immunotherapy is a promising treatment for TC, the majority of which is considered an immune-hot malignancy. Immune suppression by TC cells and immune-suppressing cells, including tumor-associated macrophages, myeloid-derived suppressor cells, and regulatory T cells, is complex and dynamic. Negative immune checkpoints, cytokines, vascular endothelial growth factors (VEGF), and indoleamine 2,3-dioxygenase 1 (IDO1) suppress antitumor T cells. Basic and translational advances in immune checkpoint inhibitors (ICIs), molecule-targeted therapy, tumor-specific immunotherapy, and their combinations have enabled us to overcome immune suppression and activate antitumor immune cells. This review summarizes current findings regarding the immune microenvironment, immunosuppression, immunological targets, and immunotherapy for TC and highlights the potential efficacy of immunotherapy.

## 1. Introduction

Thyroid cancer (TC) is a major cancer worldwide, and its incidence has consistently increased over the past 30 years partly due to the advances in diagnostic techniques [1]. TC predominantly affects women throughout the world. Despite its rising incidence, mortality from TC remains relatively low [1]. TC is classified into four categories: papillary thyroid cancer (PTC), follicular thyroid cancer (FTC), medullary thyroid cancer (MTC), and anaplastic thyroid cancer (ATC). PTC and FTC are differentiated thyroid cancers (DTC) that account for >90% of all TC [2]. Hemithyroidectomy and total thyroidectomy followed by radioiodine therapy (RAI) are widely accepted standard treatments for DTC [3]. However, the prognosis of ATC is poor, with a median overall survival (OS) of less than 7 months despite multiple treatment approaches [4]. In recent years, tyrosine kinase inhibitors (TKIs) have been applied to RAI-resistant or RAI-irresponsive TC and are effective in suppressing these diseases. However, in most cases, TC eventually develops resistance to TKI. Immunotherapy has attracted the attention of clinicians for the treatment of patients with TKI-resistant TC. Although this treatment is promising for several cancer types, its immunogenicity in thyroid cancer has not been fully investigated. ATC and half of PTC are considered immune-hot malignancies based on the NanoString platform [5], suggesting that immunotherapy is a promising treatment for advanced TC. This review explores the immunological features of advanced TC and highlights the potential efficacy of immunotherapy.

## 2. Immunity in TC Micromilieu

### 2.1. Antitumor Immune Cells

Natural killer (NK) cells account for the innate immunity and recognize tumors with reduced expression of major histocompatibility complex (MHC) class I. Natural killer group 2 member D (NKG2D) is an activation receptor in NK cells, whose ligands are MHC class I-related chains A and B (MICA/B). The tumor expression of MICA/B induced by BRAF and RAS has been observed in ATC and is responsible for the antitumor effects of NK cells [6]. Extracellular vehicles (EVs) may contribute to the cytotoxicity of NK cells in TC [7]. The infiltration of NK cells is markedly higher in PTC than in benign thyroid nodules [8], and circulating NK cells are notably increased in patients with advanced TC compared to healthy individuals [9]. In ATC, enriched CD56bright CD16^−^/low NK cells express high levels of exhaustion markers, including programmed cell death-1 (PD-1), T-cell immunoglobulin, and mucin domain 3 (TIM3), with decreased levels of NKp44, NKp30, and NKG2D [9,10,11], suggesting that the function of these NK cells is inhibited. The cytotoxicity of these NK cells is restored by PD-1 and TIM3 blockade or by neutralization of prostaglandin E2. Collectively, NK cells infiltrate TC through MICA/B, followed by NK cell exhaustion, and the dysfunction of these cells is replenished by immune checkpoint inhibitors (ICIs) or cyclooxygenase (COX) inhibitors.

Interactions between antigen-presenting cells and T cells are indispensable to activate acquired immunity. Regarding antigen-presenting cells, S100+ (mature and immature), CD1a^+^ (immature), and CD83^+^ (mature) dendritic cells (DCs) have been detected in human PTC samples [12]. Chemokine receptor-6+ DC-SIGN^+^ DCs also infiltrate PTC but are scarce in FTC [13]. DC infiltration is markedly reduced in poorly differentiated TC and ATC [14]. Although Cunha et al. reported that CD8^+^ T-cell infiltration correlates with a favorable prognosis in patients with DTC [15], the same group has shown that the combination of CD8^+^ cells and COX-2 overexpression is associated with the risk of recurrent DTC [16]. In addition to COX-2, which is considered an immunosuppressive factor that produces prostaglandin E2, the relationship between CD8^+^ T cells and TC remains to be elucidated. Similarly, the antitumor effects of B cells in TC remain unknown. Few studies have revealed that the infiltration of B cells into tumors that form tertiary lymphoid tissue is associated with a favorable prognosis in PTC [17,18]. BRAF mutations in tumors may be responsible for the reduced infiltration of B cells.

### 2.2. Immune-Suppressing Cells

Tumor-associated macrophages (TAM) and M2 macrophages suppress the expression of other immune cells. In PTC, increased TAM density is associated with extrathyroid invasion, lymph node metastasis, and tumor progression [19,20,21]. Additionally, PTC-derived TAM promotes tumor invasion and metastasis by producing CXCL8 [20]. ATC has a higher M2 macrophage (CD163^+^) infiltration than other types of cancer [22]. In addition to TAM, myeloid-derived suppressor cells (MDSCs) inhibit antitumor immune cells. The number of circulating MDSCs is associated with the aggressive characteristics of DTC [23], and the circulating MDSCs in patients with ATC are significantly higher than those in healthy individuals [24]. Mast cells are associated with angiogenesis [25], lymphangiogenesis [26], and tumor progression [27,28,29]. Melillo et al. revealed that the mast cell density in PTC was higher than that in normal tissue, and that was related to extra-thyroid tumor invasion [30]. A higher mast cell density was also found in FTC than in adenomas and was related to extracapsular extension [31]. Neutrophils support tumor growth through neutrophil extracellular traps. A high neutrophil/lymphocyte ratio is correlated with a large tumor size and a high risk of recurrence in patients with TC [32,33]. French et al. reported that the frequency of regulatory T cells (Tregs) positively correlated with lymph node metastasis in PTC [34]. They also reported that Tregs were enriched in tumor-involved lymph nodes and that their frequency was associated with PTC recurrence [35]. Liu et al. also found that a high percentage of Tregs in both peripheral blood and tumor tissue correlated with extrathyroid invasion and lymph node metastasis in PTC [36]. In addition to Tregs, regulatory B cells that inhibit IFN-γ-production from CD4^+^ and CD8^+^ T cells through IL-10 have been observed in DTC [37].

Collectively, immune-suppressive cells are frequently observed in advanced TC, and the activation of antitumor immune cells by overwhelming these suppressive cells may pave the way for establishing novel immunotherapies against TC (Figure 1).

Immune-suppressing cells, including macrophages, MDSCs, neutrophils, Tregs, and mast cells, are frequently observed in thyroid cancer. Immune-suppressing cells and tumor cells interact with each other via chemokines and cytokines.

## 3. Immunosuppression by TC

In addition to immune-suppressing cells, the tumor itself can suppress and escape antitumor immune cells, leading to tumor progression. In TC, immune escape occurs through several mechanisms. The decreased expression of MHC class I molecules and β2-microglobulin, a component of MHC, in TC cells supports the evasion of cytotoxic T cell activity by suppressing antigen presentation [38,39]. The cell signaling pathway is partly responsible for diminished antigen presentation in TC. RET is a receptor tyrosine kinase that regulates cell proliferation and survival through mitogen-activated protein kinase (MAPK) and phosphatidylinositol-3 kinase (PI3K)/Akt in sporadic MTC and some PTCs [40,41,42]. Aberrant activation of RET contributes to the reduced expression of MHC class II [43]. In addition to suppressing antigen presentation, TC upregulates negative immune checkpoint molecules, such as programmed cell death ligand-1 (PD-L1) and programmed cell death ligand-2, which suppress the activation of T and NK cells through PD-1 [44,45]. Several studies have reported that BRAF V600E mutation correlated with high levels of PD-L1 and cytotoxic T-lymphocyte-associated protein 4 (CTLA-4) [46,47,48,49,50,51]. The expression of these immune checkpoints inhibits CD8^+^ cytotoxic T cells and increases the number of FoxP3^+^ Tregs and M2 macrophages. Other negative immune checkpoints, including VISTA, B7H3, TIM3, TIGIT, LAG3, PDCD1, and PVR, have also been found in PTC, MTC, and ATC tissues [5,52,53,54].

Soluble mediators, including cytokines, chemokines, angiogenic factors, and metabolic enzymes, can diminish the anticancer effects of immune cells in TC. Both immune-suppressing and TC cells produce immune-suppressing cytokines that play crucial roles in TC development [27]. Interleukin (IL)-6 contributes to tumor cell proliferation, survival, invasion, and metastasis through MDSCs accumulation and activation. IL-6 is highly expressed in DTC and associated with tumor invasiveness [55]. ATC cells produce IL-6, which promotes tumor progression through M2 macrophage activation through signal transducers and activators of transcription (STAT) 3 signaling [56,57]. IL-10, an anti-inflammatory and immunosuppressive cytokine that contributes to immune escape by downregulating MHC class I on the cell surface, is produced by TAMs and TC cells. IL-10 expression in TC is associated with extrathyroid invasion and a large tumor size [58]. MDSCs numbers are high in patients with ATC and MTC and are associated with high IL-10 production [24]. Todaro et al. found that TC cells produce IL-4 and IL-10 that promote resistance to chemotherapy by upregulating Bcl-xL and Bcl-2, which suppress apoptosis [59,60]. Prostaglandin E2 is produced through COX2 in ATC [11]. Prostaglandin E2 suppresses the maturation and antitumor activity of NK cells against TC. Transforming growth factor (TGF)-β signaling plays different roles in cancer cells and normal cells. Exerting antimitogenic effects in normal thyroid follicular cells, TGF-β promotes cancer development, migration, invasion, and induction of epithelial–mesenchymal transition (EMT) [61,62].

Chemokines play a role in tumor growth and angiogenesis. In addition to cytokines, thyroid cells release CXC chemokines, including CXCL1, CXCL8, CXCL9, CXCL10, and CXCL11 [63]. PTC and ATC cells produce high levels of CXCL1, CXCL8, and CXCL10 [64,65]. The expression of CXCL12 was higher in PTC than in normal tissues and is associated with lymph node metastasis [66]. CXCR4 and CXCR7, both CXCL12 receptors, are highly expressed in PTC and are associated with tumor progression [67,68]. CXCL12-CXCR4 axis promotes migration, invasion, and EMT in human PTC cells through activation of the NF-κB signaling [68]. CXCR4 and CXCR7 expression is associated with large tumor size, advanced TNM staging, and short overall and recurrence-free survival in FTC [69]. Microarray analysis has revealed that CXCL8 expression is higher in ATC tissues than in normal thyroid tissues [70]. TAM may facilitate PTC metastasis through paracrine interactions with CXCL8 and CXCR1/2 [20]. CXCL8 and vascular endothelial growth factor (VEGF)-A secretion from poorly DTC is induced by thyroid-stimulating hormone (TSH) signaling, which regulates tumor angiogenesis, macrophage infiltration, and enhanced tumor growth [71]. CXCL8/CXCR1, CXCL1/CXCR2, and CXCL10/CXCR3 produced by mast cells promote TC cell proliferation, survival, invasion, EMT, and stemness [72]. Regarding the interaction between cytokines and chemokines, IFN-γ and TNF-α induce CXCL10/IP-10 production in human PTC and ATC cells [64,73].

VEGF is a key mediator of angiogenesis and an inducer of Tregs in the cancer microenvironment [74]. Both PTC and ATC express VEGF [75,76]. VEGF from TC cells induces neovascularization and suppresses DCs [77]. TC cells release VEGF-A, which recruits mast cells and correlates with the invasive tumor phenotype [30]. VEGF expression is significantly correlated with BRAF V600E expression in PTC with extrathyroid invasion [78]. VEGF-A, colony-stimulating factor 1, and CCL2 can attract monocytes to the tumor microenvironment and differentiate them into TAM [79]. Collectively, VEGF plays a significant role not only in vascularization but also in immune modulation.

Amino acid metabolites are necessary for the survival of antitumor immune cells. In addition to M2 macrophages, TC cells can produce metabolic enzymes such as indoleamine 2,3-dioxygenase 1 (IDO1) or arginase-1 (ARG1), which reduce the amino acids necessary for immune cells. In PTC, IDO1 mRNA expression is associated with tumor IDO1 immunostaining intensity and FoxP3^+^ Treg density [80]. IDO1 mRNA expression is higher in patients with ATC than in patients with PTC or MDC. TC secretes IDO, ARG-1, and TGF-β, which inhibit the expression of NK cell surface-activation receptors and decrease the number and quality of NK cells [81]. A low intratumoral CD8^+^/Foxp3^+^ ratio was observed in patients with increased expression of IDO, ARG-1, and PD-L1, which is related to the BRAF V600E mutation [47]. Taken together, these results indicate that TC can directly suppress antitumor cells through numerous pathways, including negative immune checkpoints and soluble factors.

## 4. The Immunological Targets and Immunization in TC

### 4.1. The Expression of Programmed Cell Death Ligand-1

Negative immune checkpoints expressed on the surface of immune-suppressing or tumor cells inhibit the immune system, leading to immune tolerance. The interaction between PD-L1 and PD-1 suppresses the effector functions of cytotoxic T cells and NK cells. PD-1 inhibitors have shown clinical efficacy in other cancer types, and the expression of PD-L1 is considered a favorable biomarker for PD-1 inhibitors [82,83].

Several studies have examined PD-L1 expression by immunostaining in TC cells. In DTC, the positivity of PD-L1 varied among studies. Although two studies focusing on PTC reported low PD-L1 positivity rates, ranging from 6.1 to 10.1% [53,84], other studies reported higher PD-L1 positivity rates, ranging from 0.3 to 87% [47,85,86]. It should be noted that the expression of PD-L1 is significantly higher in aggressive PTC than in non-aggressive PTC [85]. Two studies on FTC reported PD-L1 positivity rates of 7.6% and 59.7%, respectively [84,87]. PD-L1 expression is relatively low, with positivity rates ranging from 12.5 to 14.4% in MTC [88,89]. In ATC, the expression of PD-L1 is relatively high, ranging from 60 to 81.3% [53,86,90,91]. A direct comparison between histological types has shown that higher expression of PD-L1 is observed in ATC than in DTC [84,86]. PD-L1 is diffusely expressed in ATC, whereas it is localized in PTC [84]. Collectively, PD-L1 may be highly expressed in aggressive PTC and ATC, for which additional treatment is necessary. Further studies to evaluate the selection of antibodies to detect PD-L1, measurement methods such as the combined positive score [92] and cutoff values are required for the accurate detection of PD-L1.

### 4.2. The Candidates in TC-Specific Immunotherapy

The drawback of the PD-1 blockade is the non-specific activation of T cells, most of which are irrelevant to tumors and compete with anti-tumor T cells. Chimeric antigen receptor (CAR)-T therapy and cancer vaccines are promising cancer-specific immunotherapy approaches [93,94]. These immunotherapies are designed to target specific proteins expressed in the tumor cells. Thus, it is crucial to identify optimal targets that are predominantly expressed in tumors but not in normal cells [94].

Most TC and normal thyroid tissues express thyroid-specific proteins, such as thyroglobulin (TG) and TSH receptors. TG is strongly expressed in all normal thyroid samples but not in other normal tissues [95]. Patients with recurrent TC generally undergo total thyroidectomy, making these proteins ideal targets for cancer-specific immunotherapy [96]. The positivity rates for TG in the PTC, FTC, and ATC groups are 98.1%, 95.2%, and 7.5%, respectively [96,97]. In patients with DTC, the positivity rate of TSH receptors ranges from 68 to 90.8% [96,98,99], and high expression of TSH receptors in lymph node metastases is associated with poor prognosis [98,99]. Peripheral TG-reactive CD8^+^ T cells have been observed in patients with PTC [96,97,100], and TSH receptor-targeted CAR-T cell therapy has shown significant antitumor effects without apparent toxicity in vivo [96]. In FTC, calcitonin may be a target for inducing antitumor CD8^+^ T cells using a DC-based vaccine [101].

Although thyroid-specific proteins are potential targets for immunotherapy in DTC, their expression is reduced in ATC [95,102]. Tumor-associated antigens (TAAs) are proteins involved in tumor growth that are expressed at low levels in normal tissues [94]. Intercellular adhesion molecule-1 (ICAM-1) is a well-studied TAA used to treat TC. ICAM-1 is a member of the immunoglobulin superfamily that mediates cell–cell interactions, and its expression is faintly detectable in epithelial cells and normal thyroid tissues under non-inflammatory conditions. The reported positivity rate of ICAM-1 by immunostaining is 85.6% in patients with PTC [103]. Another study showed that 100% of patients with ATC were positive for ICAM-1, and the staining levels were higher in ATC than in PTC [104]. ICAM-1 expression correlates with poor prognosis and metastasis in TC [105]. The antitumor activity of ICAM-1-targeted immunotherapy with CAR-T cells and monoclonal antibody-based treatment has been reported [104,105,106,107]. ICAM-1-targeted CAR-T cell therapy has shown robust antitumor effects in PTC and ATC models [104,105,106,107].

Carcinoembryonic antigen (CEA) belongs to the immunoglobulin superfamily. CEA is mainly expressed in MTC and gastrointestinal adenocarcinoma and is associated with metastasis. Most patients with MTC express CEA, ranging from 77 to 100% [108,109,110]. Clinical trials using DCs- and yeast-based vaccines against CEA have been conducted for several types of cancers, including MTC [111,112,113]. Cancer-testis antigens are TAAs expressed only in the testis and cancer, and their expression is associated with tumor progression [94]. Among cancer-testis antigens, the positivity rate for melanoma-associated antigen (MAGE)-A3 is 94.87% in patients with DTC [114]. Milkovic et al. described that the positivity rates of MAGE-3 were 80% and 29% in PTC and FTC, respectively [115]. In TC, including poorly differentiated TC and ATC, the positivity for cancer-testis antigens is as follows: MAGE-A, 61.8%; MAGE-C1, 57.1%; G antigen, 66.7%; and cancer-testis antigen 1 B, 14.4% [116].

Tumor-specific antigens (TSAs) are highly immunogenic antigens expressed only in tumor cells. Neoantigens are TSAs produced by tumor-specific mutations and are considered suitable targets for cancer vaccines [117]. Although little is known regarding neoantigens in TC, several studies have indicated that abundant neoantigens are expressed in ATCs compared to other types of TC [118,119]. Several molecular pathways, including the MAPK and PI3K/Akt pathways, have been suggested to play roles in the development of TC [120]. Mutations in molecules within these signaling pathways, including BRAF, RAS, RET, and PTEN, have been detected in various TC types and are associated with disease progression and poor prognosis. BRAF is the most frequently mutated protein, and mutations are observed in approximately 40–45% of TC cases. The most common BRAF mutation is the V600E transversion, with some studies suggesting that 36% of patients with PTC harbor this mutation [121,122,123]. The BRAF V600E mutation activates MAPK signaling and is associated with poor outcomes and high recurrence rates [121,124]. Furthermore, 10–50% of patients with ATC carry the BRAF V600E mutation, which is associated with a poor prognosis [86,125,126,127]. Mutations or overexpression of the *RAS* gene family, including *HRAS*, *KRAS*, and *NRAS*, are observed in 20–30% of TC. RAS and RAF mutations contribute to tumor growth and survival by activating downstream signaling pathways such as MAPK/ERK and PI3K/Akt [128]. As neoantigens have been found in BRAF V600E, KRAS, and PI3K [129,130], these mutated signaling proteins are targets of TSAs in tumor-specific immunotherapy. Collectively, the activation of antigen-specific T cells in addition to NK cells is a promising strategy to treat thyroid cancer (Figure 2).

Both innate and acquired immune cells may attack thyroid cancer cells. The antitumor ability of exhausted NK cells in ATC is recovered by immune checkpoint blockade or COX inhibitor. Thyroid cancer expresses several tumor-associated antigens and neoantigens that can be recognized by antitumor T cells.

### 4.3. Immunization against TC

Active immunization (cancer vaccine) and ex vivo proliferation of antigen-specific T cells followed by adoptive cell transfer (ACT) are promising approaches to potentiate anti-TC T cells using the candidate antigens as mentioned above (e.g., TG and CEA). Although the antitumor effect of vaccination has not been thoughtfully examined in TC, this approach has achieved clinical benefits in several types of cancer. In melanoma, vaccinations or ACTs are effective to suppress tumors. Rahdan et al. have shown preventive efficacy of peptide vaccine in preclinical models [131]. The combination of peptides with appropriate adjuvants such as poly-IC and CD40 could induce robust antitumor responses in a melanoma model [132]. In castration-resistant prostate cancer, personalized peptide vaccine has achieved clinical responses in a phase 2 study [133]. In TC, a DC-based vaccine using an autologous tumor lysate as an antigen [134] and a yeast-based vaccine targeting CEA [135] have shown promising results. Further trials are necessary to confirm that immunization can induce antitumor responses in TC as well as melanoma and prostate cancer.

## 5. Experimental and Clinical Immunotherapy against TC

Owing to the challenges in establishing preclinical models, only a few studies on immunotherapy targeting TC in immunocompetent mouse models have been reported. Anti-PD-1 blockade has shown mild antitumor effects in a transgenic mouse model of spontaneous PTC [136]. In other studies, anti-PD-1 or -PD-L1 therapy had no effect on orthotopic murine ATC [51,137,138] or on a transgenic mouse model of DTC progressing to ATC [139]. Despite the weak efficacy of ICI monotherapy, combination therapy with ICIs and molecule-targeted therapies is promising for preclinical studies. Lenvatinib (an oral multi-TKI against VEGFR1-3, fibroblast growth factor receptor (FGFR)1-4, platelet-derived growth factor receptor alpha, RET, and c-kit) increases CD8^+^ T cell and cytotoxic CD4^+^ T cell infiltration with decreased polymorphonuclear MDSCs, resulting in significant antitumor effects when combined with anti-PD-1 or PD-L1 therapy against ATC [138,139]. Combination therapy with BRAF inhibitors and ICIs dramatically reduces tumor growth in PTC- and ATC-bearing mice, with increased tumor-infiltrating CD8^+^ T cells, NK cells, or CD4^+^ T cells subsequent to the upregulation of tumoral MHC class II expression through class II major histocompatibility complex transactivator [51,136,137]. Regarding tumor-specific immunotherapy, ICIs enhanced the antitumor effect of ICAM1-targeting CAR-T cell therapy in a xenograft ATC model [106]. These preclinical results suggest that immunotherapy is a promising therapeutic strategy for TC.

Clinical evidence for the efficacy of ICIs against TC has gradually emerged. In a Phase Ib clinical trial (KEYNOTE-028), 22 patients with standard treatment-resistant PTC or FTC and high PD-L1 expression were evaluated for safety and efficacy of pembrolizumab, an anti-PD-1 antibody [140]. The median progression-free survival was 7 months with a favorable safety profile, and two patients exhibited a partial response [140]. As a part of phase I/II spartalizumab study against advanced solid tumor, 42 patients with locally advanced and/or metastatic ATC were treated by PD-1 blockade in the phase II cohort [141]. The response rate was 19% (three patients with complete response and five patients with partial response). Notably, patients with PD-L1–positive tumors (PD-L1 > 50%) had high response rates (8/28; 29%), whereas patients with PD-L1-negative tumors had no responses (0/12; 0%) [141]. In another study, pembrolizumab or nivolumab showed 16% response rates with two partial response, and one-year survival rate was 38% in 13 patients with locally advanced or metastatic unresectable ATC [142]. As well as other types of tumors, such as melanoma, lung cancer, and head and neck squamous carcinoma [143,144], these results indicate that PD-1 inhibitors are effective in some patients with DTC or ATC, but the number of responders is limited. Another type of ICIs, the CTLA-4 inhibitor, is considered to activate T-cell priming. Although it is difficult to interpret the results of a phase I trial, three patients with TC who received CTLA-4 inhibitor monotherapy did not show significant antitumor effects [145].

Due to the unsatisfactory results of monotherapy in clinical trials, combined approaches using ICIs have been explored. Xing et al. reported that patients with PD-L1 positive ATC achieved a complete response to a combination of radiotherapy and an anti-PD-1 antibody (tislelizumab) [146]. Although chemotherapy combined with ICIs has shown favorable results in some cancer types, a phase II clinical trial of pembrolizumab with chemoradiotherapy (docetaxel/doxorubicin) showed no clinical benefit in three patients with ATC [147]. To examine the use of ICIs in TC, an ongoing clinical trial is evaluating the clinical benefits of combining ipilimumab and nivolumab in radioiodine-refractory TC (NCT03246958).

Small-molecule-targeted therapy synergizes with ICIs by modulating the immune microenvironment. Similar to preclinical studies, clinical cases have shown favorable responses to ICIs combined with targeted therapies, including lenvatinib [148,149,150,151,152]. These studies showed that the combination therapy against ATC achieved 60–66% overall response rates, and the median overall survival was 8.3–18.5 months [148,149]. As several prospective studies have reported that the monotherapy of lenvatinib against ATC showed response rates of 3–24%, and the median overall survival ranged from 3.2 to 10.6 months [153,154,155,156], this combined approach would be a hopeful treatment. Other multi-TKIs, such as cabozantinib and anlotinib, have also shown synergy with anti-PD-1 therapy, leading to long-term survival in patients with ATC [157,158]. Moreover, ICIs combined with apatinib, a TKI that selectively inhibits VEGFR-2, achieved a partial response in a patient with radioiodine-refractory DTC [159]. Dabrafenib and trametinib (BRAF and MEK inhibitors) have shown a complete response against ATC with ICIs [160,161]. To date, the inhibition of MEK, FGFR, and VEGFR has been shown to modulate immunity by upregulating MHC expression and activating T cells in other tumor types [162,163,164]. As these molecules are widely expressed in TC, a more detailed immunological analysis of these proteins in TC is required to integrate basic and translational studies into clinical practice.

## 6. Conclusions and Future Directions

In this review, we summarize current findings on the immune microenvironment, immunosuppression, immunological targets, and immunotherapy in advanced TC. According to the immune surveillance hypothesis, aggressive tumors often escape antitumor immunity. Tumor MHC expression and CD8^+^ T cell and NK cell infiltration are suppressed in advanced TC [15,165]. Although TAM, MDSCs, Tregs, negative immune checkpoints, cytokines/chemokines, VEGF, and IDO-1 are hypothesized to suppress antitumor T cells in aggressive TC, ICIs and VEGF are the only actionable targets to date. As molecule-targeted therapies such as MAPK inhibitors upregulate the cytotoxicity of immune cells [6], the combination of ICIs and molecule-targeted therapy is a promising approach for activating antitumor immunity (Figure 3). Although the combination of PD-1 blockades and lenvatinib has shown promising clinical responses as a first-line therapy for advanced or recurrent endometrial cancer [166], LEAP-08 and LEAP-10 did not show clinical benefits in lung cancer and head and neck cancer, respectively. This combination should also be tested for TC, for which lenvatinib has already been clinically approved. Further prospective clinical trials combining molecular targeted therapy with ICIs are required to treat TC.

Despite the release of T cells from negative immune checkpoints activating antitumor T cells, only a limited number of patients respond to tumor-nonspecific immunotherapy. Patients with enriched inflammatory response pathways and high levels of immune cell infiltration may respond to ICIs in TC [167]; however, particular biomarkers to identify ICI responders are yet to be identified. PD-L1 expression in the stroma or the detection of tumor-reactive T cells using a TAA-derived epitope would be useful for determining responders to ICIs. Further treatment options should be established for most patients with TC who cannot respond to ICIs. As tumor-specific immunotherapy has exhibited significant antitumor effects in preclinical PTC and ATC models [105,106], further translational research is necessary to confirm the clinical responses to this novel mode of immunotherapy in advanced TC.

Various molecules and pathways, including decreased MHC and increased negative immune checkpoints such as PDL1, contribute to immune escape of thyroid cancer. The inhibitors of signaling pathways (e.g., EGFR, RET, and VEGFR) can be used as immunomodulators. The antigens derived from thyroid cancer may be a source of cancer vaccines and CAR-T therapy. TAA: tumor-associated antigen.

## Figures and Tables

**Figure 1 vaccines-12-00045-f001:**
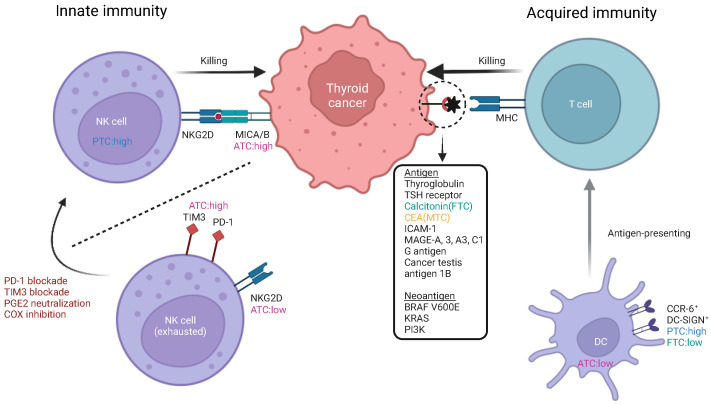
Interaction of immune-suppressing cells with thyroid cancer.

**Figure 2 vaccines-12-00045-f002:**
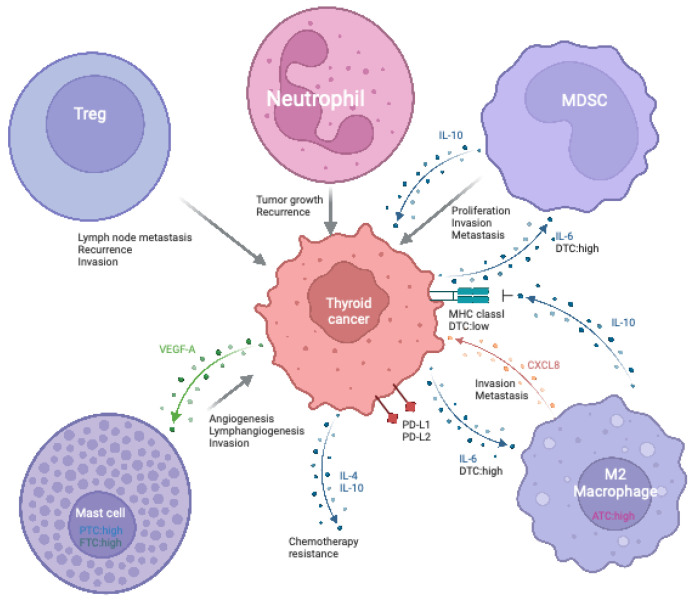
Antitumor immunity in thyroid cancer.

**Figure 3 vaccines-12-00045-f003:**
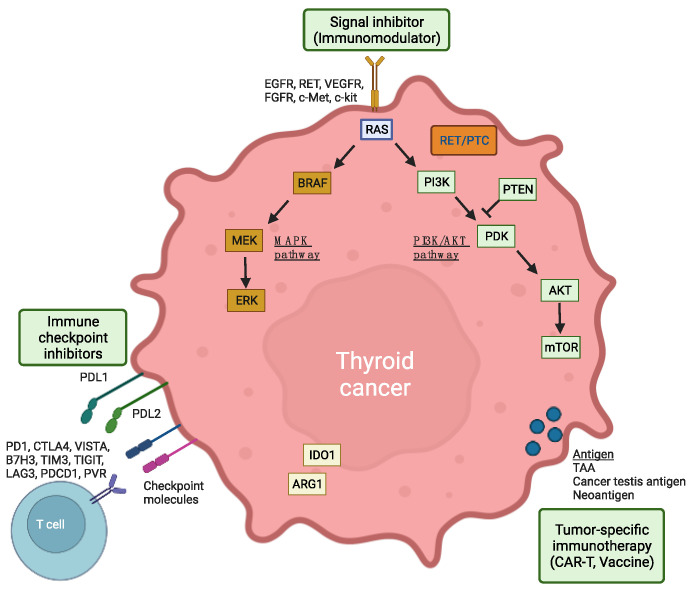
Immunotherapeutic Targets for Thyroid Cancer.

## Data Availability

No new data were created or analyzed in this study. Data sharing is not applicable to this article.

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
