# Peer review of "Harnessing Immunity to Treat Advanced Thyroid Cancer"

_vaccines, 2023, doi:10.3390/vaccines12010045_

Round 1

Reviewer 1 Report

Comments and Suggestions for Authors

The authors did a good job explaining in detail the mechanisms by which thyroid cancer is threatened by the immune system, but also the mechanisms by which the tumor evaids these mechanisms. However, I do not think the manuscript reaches the scope of the journal “vaccines” in the present form. At some points the manuscript focuses on immunotherapy, but not immunization against cancer. The authors should (add the section) reflect on how these described mechanisms could be of use for preventing the disease onset, and which delivery mechanisms could be employed.

Author Response

Reviewer #1: The authors did a good job explaining in detail the mechanisms by which thyroid cancer is threatened by the immune system, but also the mechanisms by which the tumor evades these mechanisms. However, I do not think the manuscript reaches the scope of the journal “vaccines” in the present form. At some points the manuscript focuses on immunotherapy, but not immunization against cancer. The authors should (add the section) reflect on how these described mechanisms could be of use for preventing the disease onset, and which delivery mechanisms could be employed.

>Answer: We thank the reviewer for considering our article valuable and interesting. We have added a chapter regarding immunization as suggested (p.7 lines 289-302).

Reviewer 2 Report

Comments and Suggestions for Authors

- Minor editing required (punctuation), please check.

- Attached file show some observations and suggestions.

* Blue: Format.

* Yellow: Changes suggested.

Comments on the Quality of English Language

I consider the quality of the English language is acceptable, minor suggestions on my part are mentioned above and highlighted in the attached document.

Author Response

Reviewer #2: - Minor editing required (punctuation), please check.

- Attached file show some observations and suggestions.

* Blue: Format.

* Yellow: Changes suggested.

I consider the quality of the English language is acceptable, minor suggestions on my part are mentioned above and highlighted in the attached document.

>Answer: We appreciate the thorough review and all the comments from this reviewer, which have further improved the quality of this paper. We have corrected all the grammar and superscripts as suggested, and the revised manuscript was proofread by native speakers (Editage®). Thank you for reviewing and considering our manuscript interesting.

Reviewer 3 Report

Comments and Suggestions for Authors

This manuscript presents a helpful review of the preclinical and limited clinical data to support a role for immunotherapy in thyroid cancer. This is very topical, in an era where immunotherapy is playing an increasing role in the management of many cancers.

I wonder if a slight amendment of the title to 'Harnessing immunity to treat advanced thyroid cancer' would be helpful- to emphasise that this really only applies to a minority of thyroid cancer cases.

The abstract provides a helpful summary of the issues discussed in the paper. There are however a couple of inaccurate statements which should be corrected.

At line 20 it is stated that there is no effective therapy for anaplastic thyroid cancer. This is no longer strictly true, given the emergence of Dabrafenib and Trametinib as an active treatment option for patients with BRAF mutated tumours which has made a significant difference to outcomes. 

At line 23 it is stated that thyroid cancer is an immune-hot malignancy. This statement requires some qualification. Whilst a proportion of thyroid cancers are 'hot' this does not apply to all thyroid cancers and indeed there is very much a range- as described in Riccardo Giannini, Sonia Moretti, Clara Ugolini et al. Immune Profiling of Thyroid Carcinomas Suggests the Existence of Two Major Phenotypes: An ATC-Like and a PDTC-Like, The Journal of Clinical Endocrinology & Metabolism, Volume 104, Issue 8, August 2019, Pages 3557–3575, https://doi.org/10.1210/jc.2018-01167 referenced later in the paper.

The introduction provides useful context and background information. Whilst it is true that thyroid cancer incidence has increased, it could perhaps be made clearer that this is generally due to increased diagnosis of low risk, excellent prognosis cancers (hence lack of increase in mortality), which are really not relevant to the discussion in this paper. It could perhaps be made clearer that the consideration of immunotherapy in this paper really only applies to a very small proportion of thyroid cancers which are advanced and carry a poor prognosis.

At lines 40-41, differentiated thyroid cancer is defined as a combination of PTC, FTC and MTC. This is not strictly true- DTC is a label usually applied to PTC and FTC and NOT MTC- and certainly radioiodine therapy would not be used for MTC.

There follow a number of sections reviewing the largely pre-clinical data to support a role for immune targeted therapy in thyroid cancer. As a general comment, whilst it is implied in the discussion, I think it would be helpful to make it clearer that most of this data relates to advanced, aggressive thyroid cancers such as ATC, and is probably not applicable to the majority of low risk disease.

The text is accompanied by helpful figures which illustrate the points made very well.

These sections conclude with a helpful review of the currently limited available clinical data. A couple of points could be clarified here. The sentence at lines 315-316 requires review as it states that 'PD-LI positive tumour had high response rates.... but no response rates for PD-L1>50% (??should be PD-L1<50%?) and PD-L1 negative tumours'. 

At lines 323-324 it is stated that a CTLA-4 inhibitor did not show significant clinical activity in a phase 1 trial including 3 patients. This statement should perhaps be qualified by stating that phase 1 trials are not designed to evaluate efficacy, and a sample of 3 patients is really too small to draw any conclusions about efficacy.

The conclusions drawn in the final section appear reasonable, but again, it might be made clearer that this approach is really only relevant to the small minority of aggressive, poor prognosis thyroid cancer, rather than being applicable to all thyroid cancers.

The references included all appear to be appropriate.

Author Response

Reviewer #3: This manuscript presents a helpful review of the preclinical and limited clinical data to support a role for immunotherapy in thyroid cancer. This is very topical, in an era where immunotherapy is playing an increasing role in the management of many cancers.

>Answer: We thank this reviewer for reviewing and considering our manuscript interesting. We appreciate the comments from this reviewer, which have further improved the merits of this paper.

  1. I wonder if a slight amendment of the title to 'Harnessing immunity to treat advanced thyroid cancer' would be helpful- to emphasise that this really only applies to a minority of thyroid cancer cases.

>Answer: We have revised the title as suggested.

  1. The abstract provides a helpful summary of the issues discussed in the paper. There are however a couple of inaccurate statements which should be corrected.

At line 20 it is stated that there is no effective therapy for anaplastic thyroid cancer. This is no longer strictly true, given the emergence of Dabrafenib and Trametinib as an active treatment option for patients with BRAF mutated tumours which has made a significant difference to outcomes.

At line 23 it is stated that thyroid cancer is an immune-hot malignancy. This statement requires some qualification. Whilst a proportion of thyroid cancers are 'hot' this does not apply to all thyroid cancers and indeed there is very much a range- as described in Riccardo Giannini, Sonia Moretti, Clara Ugolini et al. Immune Profiling of Thyroid Carcinomas Suggests the Existence of Two Major Phenotypes: An ATC-Like and a PDTC-Like, The Journal of Clinical Endocrinology & Metabolism, Volume 104, Issue 8, August 2019, Pages 3557–3575, https://doi.org/10.1210/jc.2018-01167 referenced later in the paper.

>Answer: We apologize for the incorrect information, and have revised the manuscript (p.1 lines20 and 23).

  1. The introduction provides useful context and background information. Whilst it is true that thyroid cancer incidence has increased, it could perhaps be made clearer that this is generally due to increased diagnosis of low risk, excellent prognosis cancers (hence lack of increase in mortality), which are really not relevant to the discussion in this paper. It could perhaps be made clearer that the consideration of immunotherapy in this paper really only applies to a very small proportion of thyroid cancers which are advanced and carry a poor prognosis.

At lines 40-41, differentiated thyroid cancer is defined as a combination of PTC, FTC and MTC. This is not strictly true- DTC is a label usually applied to PTC and FTC and NOT MTC- and certainly radioiodine therapy would not be used for MTC.

>Answer: We agree with the reviewer that the advance in diagnostic techniques might be a cause of increased thyroid cancer incidence. In addition, DTC does not include FTC as suggested. We have revised the manuscript to reflect these changes (p.1-2 lines 37, 41, 52, and 53).

  1. There follow a number of sections reviewing the largely pre-clinical data to support a role for immune targeted therapy in thyroid cancer. As a general comment, whilst it is implied in the discussion, I think it would be helpful to make it clearer that most of this data relates to advanced, aggressive thyroid cancers such as ATC, and is probably not applicable to the majority of low risk disease.

>Answer: We thank the reviewer for raising this issue. We have revised “TC” to “advanced TC” throughout the article to avoid confusion.

  1. The text is accompanied by helpful figures which illustrate the points made very well.

These sections conclude with a helpful review of the currently limited available clinical data. A couple of points could be clarified here. The sentence at lines 315-316 requires review as it states that 'PD-LI positive tumour had high response rates.... but no response rates for PD-L1>50% (??should be PD-L1<50%?) and PD-L1 negative tumours'.

>Answer: We apologize for the confusion, the patient with high PD-L1 expression had high clinical responses as suggested. We have corrected the sentence in the revised manuscript (p.8 line 332).

  1. At lines 323-324 it is stated that a CTLA-4 inhibitor did not show significant clinical activity in a phase 1 trial including 3 patients. This statement should perhaps be qualified by stating that phase 1 trials are not designed to evaluate efficacy, and a sample of 3 patients is really too small to draw any conclusions about efficacy.

>Answer: We agree with the reviewer, and have revised the text in the manuscript (p.8 lines 339-340).

  1. The conclusions drawn in the final section appear reasonable, but again, it might be made clearer that this approach is really only relevant to the small minority of aggressive, poor prognosis thyroid cancer, rather than being applicable to all thyroid cancers.

The references included all appear to be appropriate.

>Answer: We agree with the reviewer that only a small population may receive benefits from immunotherapy. We have revised the manuscript (p.9 line 371 and 395).

Round 2

Reviewer 1 Report

Comments and Suggestions for Authors

The manuscript has been changed according to the suggestion, so it could be accepted for publication.